# ADAPTING AUXILIARY LOSSES USING GRADIENT SIMILARITY

## ABSTRACT

One approach to deal with the statistical inefficiency of neural networks is to rely on auxiliary losses that help to build useful representations. However, it is not always trivial to know if an auxiliary task will be helpful for the main task and when it could start hurting. We propose to use the cosine similarity between gradients of tasks as an adaptive weight to detect when an auxiliary loss is helpful to the main loss. We show that our approach is guaranteed to converge to critical points of the main task and demonstrate the practical usefulness of the proposed algorithm in a few domains: multi-task supervised learning on subsets of ImageNet, reinforcement learning on gridworld, and reinforcement learning on Atari games.

## 1 INTRODUCTION

Neural networks are extremely powerful function approximators that have excelled on a wide range of tasks (Simonyan and Zisserman, 2015; Mnih et al., 2015; He et al., 2016a; Silver et al., 2016; Vaswani et al., 2017). Despite the state of the art results across domains, they remain data-inefficient and expensive to train. In supervised learning (e.g., image classification), large deep learning (DL) benchmarks with millions of examples are needed for training (Russakovsky et al., 2015) and the additional implication of requiring human intervention to label a large dataset can be prohibitively expensive. In reinforcement learning (RL), agents typically consume millions of frames of experiences before learning to act in complex environments (Silver et al., 2016; Espeholt et al., 2018), which not only puts pressure on compute power but also makes particular domains (e.g., robotics) impractical.

Different techniques have been studied for improving data efficiency, from data augmentation (Krizhevsky et al., 2012; Simonyan and Zisserman, 2015; Hauberg et al., 2016) to transfer learning (Taylor and Stone, 2009; Pan et al., 2010). In this work, we focus on a particular setup for transfer learning. We assume that besides the main task, one has access to one or more auxiliary tasks that share some unknown structure with the main task. To improve data efficiency, these additional tasks can be used as auxiliary losses. However, only the performance on the main task is of interest, even though the model is trained simultaneously on all these tasks. Any improvement on the auxiliary losses is useful only to the extent that it helps learning features or behaviors for the main task.

Auxiliary tasks have been shown to work well in practice (e.g., Zhang et al., 2016; Jaderberg et al., 2017; Mirowski et al., 2017; Papoudakis et al., 2018). However, their success depends on how well aligned the auxiliary losses are with the main task. Knowing this apriori is typically non-trivial and the usefulness of an auxiliary task can change through the course of training. In this work, we explore a simple yet effective approach for measuring the similarity between an auxiliary task and the main task of interest, given the value of their parameters. We show that this measure can be used to decide which auxiliary losses are helpful and for how long.

### 1.1 NOTATION AND PROBLEM DESCRIPTION

Assume we have a main task $\mathcal{T}_{main}$ and an auxiliary task $\mathcal{T}_{aux}$ that induce two losses $\mathcal{L}_{main}$ and $\mathcal{L}_{aux}$. We care about only about maximizing performance on $\mathcal{T}_{main}$; $\mathcal{T}_{aux}$ is an auxiliary task which is not of direct interest. *The goal is to devise an algorithm that can automatically (i) leverage $\mathcal{T}_{aux}$ when it is helpful (e.g. learn faster) and (ii) block negative transfer when $\mathcal{T}_{aux}$ is not helpful (i.e. recover the performance of training only on $\mathcal{T}_{main}$).* Note that this setup is different from multi-objective optimization in which both the tasks are of interest. We propose to parameterize the solution for

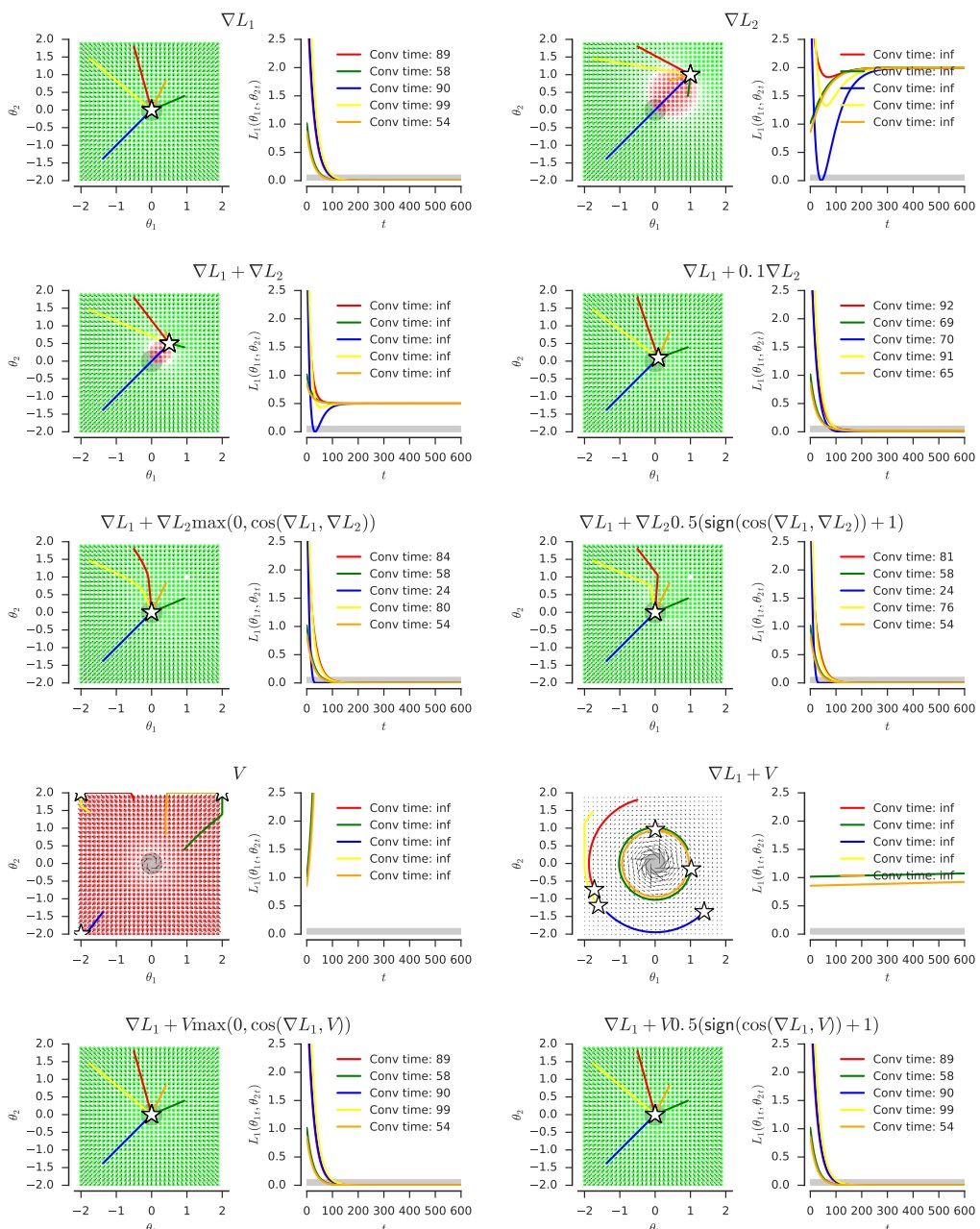

Figure 1: Positive example optimization for $L_1(\theta_1, \theta_2) = \theta_1^2 + \theta_2^2$, $L_2(\theta_1, \theta_2) = (\theta_1 - 1)^2 + (\theta_2 - 1)^2$ and $V(\theta_1, \theta_2) = [-\frac{\theta_2}{\theta_1^2 + \theta_2^2} - 2\theta_1, \frac{\theta_1}{\theta_1^2 + \theta_2^2} - 2\theta_2]$ where the proposed method speeds up the process (compared on all runs). Each colored trajectory represents one optimization run with random initial position. Star represents the convergence point. All experiments use steepest descent method and run 600 iterations with a constant step size of 0.01. Convergence time is defined as number of steps needed to get below 0.1 loss of $L_1$ (gray region). Color of each point represents its alignment with $\nabla L_1$ (green—positive alignment, red—negative alignment, white—directions are perpendicular). In this example $L_2$ is helpful for $L_1$ as it reinforces good descent directions in most of the space. However, simple mixing is actually slowing optimization down (or makes it fail completely, see the second row), while the proposed methods (weighted and unweighted variants) converge faster (see the third row). When using non-conservative vector field $V$ one obtains lack of convergence (cyclic behaviour, see the fourth row), while the proposed merging still works well (see the last row).

$\mathcal{T}_{main}$ and $\mathcal{T}_{aux}$ by two neural networks, $f(\cdot, \boldsymbol{\theta}, \boldsymbol{\phi}_{main})$ and $g(\cdot, \boldsymbol{\theta}, \boldsymbol{\phi}_{aux})$, such that they share a subset of parameters denoted here by $\boldsymbol{\theta}$. Generally, the auxiliary loss literature proposes to minimize

$$\underset{\boldsymbol{\theta}, \boldsymbol{\phi}_{main}, \boldsymbol{\phi}_{aux}}{\arg\min} \quad \mathcal{L}_{main}(\boldsymbol{\theta}, \boldsymbol{\phi}_{main}) + \lambda \mathcal{L}_{aux}(\boldsymbol{\theta}, \boldsymbol{\phi}_{aux}) \tag{1}$$

under the intuition that modifying $\boldsymbol{\theta}$ to minimize $\mathcal{L}_{aux}$ will improve $\mathcal{L}_{main}$ if the two tasks are sufficiently related. We propose to modulate the weight $\lambda$ at each learning iteration $t$ by how useful $\mathcal{T}_{aux}$ is for $\mathcal{T}_{main}$ given $\boldsymbol{\theta}^{(t)}, \boldsymbol{\phi}_{main}^{(t)}, \boldsymbol{\phi}_{aux}^{(t)}$. That is, at each optimization iteration, we want to efficiently approximate the solution to

$$\underset{\lambda^{(t)}}{\arg\min} \, \mathcal{L}_{main} \left( \boldsymbol{\theta}^{(t)} - \alpha \nabla_{\boldsymbol{\theta}}(\mathcal{L}_{main} + \lambda^{(t)} \mathcal{L}_{aux}), \boldsymbol{\phi}_{main}^{(t)} - \alpha \nabla_{\boldsymbol{\phi}_{main}} \mathcal{L}_{main} \right). \tag{2}$$

Note that the input space of $\mathcal{T}_{main}$ and $\mathcal{T}_{aux}$ do not have to match. In particular, $\mathcal{T}_{aux}$ does not need to be defined for an input of $\mathcal{T}_{main}$ or the other way around.[1] Solving equation 2 is expensive. Instead, we look for a cheap heuristic to approximate $\lambda^{(t)}$ which is better than keeping $\lambda^{(t)}$ constant and does not require hyper-tuning.

## 2    COSINE SIMILARITY BETWEEN GRADIENTS OF TASKS

We propose to use the cosine similarity of gradients between tasks as a measure of task similarity and hence for approximating $\lambda^{(t)}$. Consider an example where the main function to minimize is $\mathcal{L}_{main} = (\theta - 10)^2$ and the auxiliary function is $\mathcal{L}_{aux} = \theta^2$, their gradients are $\nabla_\theta \mathcal{L}_{main} = 2(\theta - 10)$ and $\nabla_\theta \mathcal{L}_{aux} = 2\theta$ respectively. When $\theta$ is initialized at $\theta = -20$, the gradients of the main and auxiliary functions point in the same direction and the cosine similarity is 1; minimizing the auxiliary loss is beneficial for minimizing the main. However, at a different point, $\theta = 5$, the two gradients point in different directions and the cosine similarity is $-1$; minimizing the auxiliary loss would hinder minimizing the main loss (See Figure 7 in Appendix B for an illustration.).

This example suggests a natural strategy for approximating $\lambda^{(t)}$: *minimize the auxiliary loss as long as its gradient has non-negative cosine similarity with the target gradient; otherwise, the auxiliary loss should be ignored.* This follows the well-known intuition that if a vector is in the same half-space as the gradient of a function $f$, then it is a *decent direction* for $f$. This reduces our strategy to ask if the gradient of the auxiliary loss is a *descent direction* for the main loss of interest.

**Proposition 1.** *Given any gradient vector field $G(\boldsymbol{\theta}) = \nabla_{\boldsymbol{\theta}} \mathcal{L}(\boldsymbol{\theta})$ and any vector field $V(\boldsymbol{\theta})$ (such as the gradient of another loss function, or an arbitrary set of updates), an update rule of the form*

$$\boldsymbol{\theta}^{(t+1)} := \boldsymbol{\theta}^{(t)} - \alpha^{(t)}(G(\boldsymbol{\theta}^{(t)}) + V(\boldsymbol{\theta}^{(t)}) \max(0, \cos(G(\boldsymbol{\theta}^{(t)}), V(\boldsymbol{\theta}^{(t)}))))$$

*converges to the local minimum of $\mathcal{L}$ given small enough $\alpha^{(t)}$.*

Proof is provided in Appendix A.1.

Note that the above statement does not guarantee any improvement of convergence, but only guarantees lack of divergence. In particular, cosine similarity is not a silver bullet that guarantees positive transfer, but it can drop the "worst-case scenarios". In principle, one can create example functions where the convergence of the main loss is affected both positively (see Figure 1) and negatively (see Figure 8 in Appendix D). Nevertheless, convergence on the main task is guaranteed for our proposed strategy, as the proposition shows.

In addition, it is important to note that simply adding an arbitrary vector field does not have the convergence property. For example, use function $V(\boldsymbol{\theta}) = -\nabla_{\boldsymbol{\theta}} \mathcal{L}(\boldsymbol{\theta}) + \left[ -\frac{\theta_2}{\theta_1^2 + \theta_2^2}, \frac{\theta_1}{\theta_1^2 + \theta_2^2} \right]^T$ as a two-dimensional case, which leads to an update rule of $\boldsymbol{\theta}^{(t+1)} = \boldsymbol{\theta}^{(t)} - \alpha \left[ -\frac{\theta_2}{\theta_1^2 + \theta_2^2}, \frac{\theta_1}{\theta_1^2 + \theta_2^2} \right]^T$. This is a non-conservative vector field which cases the optimizer to follow concentric circles around the origin (see the fourth row in Figure 1). This is crucial to note for some realistic scenarios where one does not always form a gradient field (e.g., the update rule of the Q-learning algorithm in RL).

---

[1] In the supervised learning case when the input features are shared, this setting resembles the *multi-task learning without label correspondences* setting (Quadrianto et al., 2010).

Figure 1 provides a few illustrative examples on quadratic functions using the proposed approach, which helps intuitively understand the kind of scenarios for which the approach could help.

The above proposition refers to losses with the same set of parameters $\boldsymbol{\theta}$, while equation 2 refers to the scenario when each loss has task specific parameters (e.g. $\phi_{main}$ and $\phi_{aux}$). The following proposition extends to this scenario:

**Proposition 2.** *Given two losses parametrized with $\boldsymbol{\Theta}$ (some of which are shared $\boldsymbol{\theta}$ and some unique to each loss $\phi_{main}$ and $\phi_{aux}$), learning rule:*

$$\boldsymbol{\theta}^{(t+1)} := \boldsymbol{\theta}^{(t)} - \alpha^{(t)}(\nabla_{\boldsymbol{\theta}}\mathcal{L}_{main}(\boldsymbol{\theta}^{(t)}) + \nabla_{\boldsymbol{\theta}}\mathcal{L}_{aux}(\boldsymbol{\theta}^{(t)})\max(0,\cos(\nabla_{\boldsymbol{\theta}}\mathcal{L}_{main}(\boldsymbol{\theta}^{(t)}),\nabla_{\boldsymbol{\theta}}\mathcal{L}_{aux}(\boldsymbol{\theta}^{(t)}))))$$

$$\phi_{main}^{(t+1)} := \phi_{main}^{(t)} - \alpha^{(t)}\nabla_{\phi_{main}}\mathcal{L}_{main}(\boldsymbol{\Theta}^{(t)}) \quad and \quad \phi_{aux}^{(t+1)} := \phi_{aux}^{(t)} - \alpha^{(t)}\nabla_{\phi_{aux}}\mathcal{L}_{aux}(\boldsymbol{\Theta}^{(t)})$$

*leads to convergence to local minimum of $\mathcal{L}_{main}$ w.r.t. $(\boldsymbol{\theta},\phi_{main})$ given small enough $\alpha^{(t)}$.*

*Proof.* Comes directly from the previous proposition that $G = \nabla_{\boldsymbol{\theta}}\mathcal{L}_{main}$ and $V = \nabla_{\boldsymbol{\theta}}\mathcal{L}_{aux}$. For any vector fields $A, B, C$, we have $\langle A, B\rangle \geq 0$ and $\langle C, B\rangle \geq 0$ implies $\langle A + C, B\rangle \geq 0$. □

Analogous guarantees hold for the *unweighted* version of this algorithm, where instead of weighting by $\cos(G, V)$ we use a binary weight $(\text{sign}(\cos(G, V)) + 1)/2$ which is equivalent to using $V$ iff $\cos(G, V) > 0$. When training with mini-batches, accurately estimating $\cos(G, V)$ can be difficult due to mini-batch noise; the unweighted variant only requires $\text{sign}(\cos(G, V))$ which can be estimated more robustly. Hence, we use this variant in our experiments unless otherwise specified. Additionally, note that there is no guarantee that $\mathcal{L}_{aux}$ is optimized. For example, if $\mathcal{L}_{aux} = -\mathcal{L}_{main}$ then $\mathcal{L}_{aux}$ is ignored (see a visualization in the last row of Figure 1).

Despite its simplicity, the proposed update rule can give rise to interesting phenomena. We can show that the emerging vector field could be non-conservative, which means there does not exist a loss function for which it is a gradient. While this might seem problematic (for gradient-descent-based optimizers), it describes only the global structure—typically used optimizers are local in nature and they do local, linear or quadratic approximations of the function (Shwartz-Ziv and Tishby, 2017). Consequently, in practice, one should not expect any negative effects from this phenomena, as it simply shows that our proposed technique is in fact qualitatively changing the nature of the update rules for training.

**Proposition 3.** *In general, the proposed update rule does not have to create a conservative vector field.*

Proof is provided in Appendix A.2.

## 3 APPLICATIONS OF GRADIENT COSINE SIMILARITY

In this section, we use the gradient cosine similarity to decide when to train on the auxiliary task. All experiments (unless otherwise stated) follow the *unweighted* version of our method, summarized in Algorithm 1. The *weighted* version of our method is summarized in Algorithm 2, Appendix C.

---
**Algorithm 1** Unweighted version of our method.

1: Initialize shared parameters $\boldsymbol{\theta}$ and task specific parameters $\phi_{main}, \phi_{aux}$ randomly.
2: **for** iter $= 1 :$ max_iter **do**
3:    Compute $\nabla_{\boldsymbol{\theta}}\mathcal{L}_{main}, \nabla_{\phi_{main}}\mathcal{L}_{main}, \nabla_{\boldsymbol{\theta}}\mathcal{L}_{aux}, \nabla_{\phi_{aux}}\mathcal{L}_{aux}$.
4:    Update $\phi_{main}$ and $\phi_{aux}$ using corresponding gradients
5:    **if** $\cos(\nabla_{\boldsymbol{\theta}}\mathcal{L}_{main}, \nabla_{\boldsymbol{\theta}}\mathcal{L}_{aux}) \geq 0$ **then**
6:      Update $\boldsymbol{\theta}$ using $\nabla_{\boldsymbol{\theta}}\mathcal{L}_{main} + \nabla_{\boldsymbol{\theta}}\mathcal{L}_{aux}$
7:    **else**
8:      Update $\boldsymbol{\theta}$ using $\nabla_{\boldsymbol{\theta}}\mathcal{L}_{main}$
---

### 3.1 EXPERIMENTS ON IMAGE CLASSIFICATION TASKS

First, we consider a classification problem on ImageNet (Russakovsky et al., 2015) and design a simple multi-task binary classification task to test our hypothesis that *transferable tasks should have high cosine similarity (and vice versa)*. We take a pair of classes from ImageNet, refer to these as

class $A$ and class $B$; all the other 998 classes in ImageNet (except $A$ and $B$) are referred to as the *background*. Our tasks $\mathcal{T}_{main}$ and $\mathcal{T}_{aux}$ are then formed as a binary classification of *if an image is class A (otherwise background)* and *if an image is class B (otherwise background)* respectively.

Ideally, we want to pick groups of class pairs that reflect *near* or *far* distance, for the purpose of providing a baseline of *transferability*. Therefore, we used two distance measures, *lowest common ancestor (LCA)* in the ImageNet label hierarchy and *Frechet Inception Distance (FID)* (Heusel et al., 2017) of pre-trained image embedding, to serve as a ground truth of class similarity for selecting class pair $A$ and $B$. Based on these measures, we picked three pair of classes for *near*, class 871 (trimaran) vs. 484 (catamaran), 250 (Siberian husky) vs. 249 (malamute), and 238 (Greater Swiss Mountain dog) vs. 241 (Entleucher); and for *far*, class 920 (traffic light) vs. 62 (rock python), 926 (hotpot) vs. 800 (slot), and 48 (Komodo dragon) vs. 920 (traffic light). Details on the class pair selection are described in Appendix E.

We use a modified ResNetV2-18 model (He et al., 2016b) for training in this experiment. All parameters in the convolutional layers are shared (denote as $\boldsymbol{\theta}$), followed by task-specific parameters $\phi_{main}$ and $\phi_{aux}$. First, we use a multi-task learning setup and minimize $\mathcal{L}_{main} + \mathcal{L}_{aux}$, and measure cosine similarity on $\boldsymbol{\theta}$ through the course of training. Figure 2(a) shows that cosine similarity is higher for *near* pairs (blue lines) and lower for *far* pairs (red lines). Next, we compare single-task training, multi-task training, and our proposed variant on two scenarios (i) auxiliary task helps and (ii) auxiliary task hurts. As mentioned earlier, our goal is have a method that can automatically leverage auxiliary tasks when they are helpful and avoid negative transfer when auxiliary tasks are not helpful. Figure 2(b) shows that on a *near* pair, all variants perform similarly in terms of final performance; furthermore, our method performs similar to multi-task learning and learns faster than single task because the task is transferable. Figure 2(c) shows that on a *far* pair, multi-task learning leads to poorer performance than single-task learning on the main task due to the potential negative transfer, whereas our method of using gradient cosine similarity blocks negative transfer and automatically achieves performance that is comparable to single-task learning.

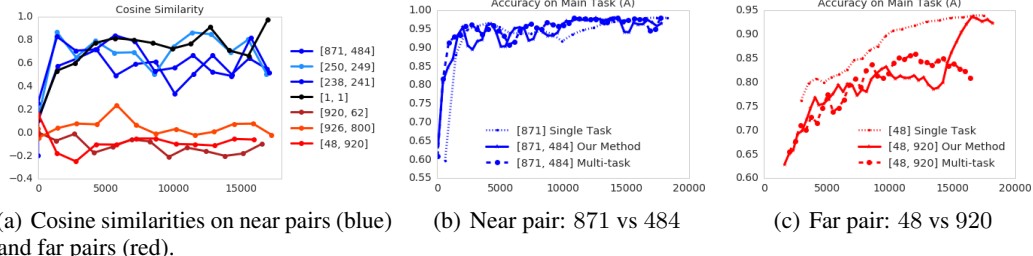

(a) Cosine similarities on near pairs (blue) and far pairs (red).

(b) Near pair: 871 vs 484

(c) Far pair: 48 vs 920

Figure 2: Multi-task learning setup on ImageNet class pairs. *(a)*: gradient cosine similarity is higher for near pairs and lower for far pairs. *(b)* and *(c)*: testing accuracy on single task (dotted), naive multi-task (dashed), and our method (solid). Naive multi-task learning helps in near pairs (see *(b)*) but hurts in far pairs (see *(c)*) because of its lack of the ability to prevent negative effects from the auxiliary task to the main task. Our method can overcome this limitation by dropping the auxiliary task when its gradient direction disagrees with the main task, thus achieving the best of both worlds: matching the multi-task performance on near pairs (see *(b)* where our method and multi-task learning learn faster than single task only) and the single task performance in far pairs (see *(c)* where multi-task learning performs poorly, but our method automatically recovers single task performance).

## 3.2 EXPERIMENTS ON REINFORCEMENT LEARNING GRIDWORLD TASKS

We then consider a typical RL problem where one aims to find a policy $\pi$ that maximizes sum of future discounted rewards $\mathbb{E}_\pi[\sum_{t'=1}^{N}\gamma^{t'-1}r_{t'}]$ in a partially observable Markov decision process (POMDP). There have been many techniques proposed to solve this optimization problem, from classical policy gradient (Williams, 1992), Q-Learning (Watkins, 1989), to the more modern Proximal Policy Optimization (Schulman et al., 2017) and V-Trace (Espeholt et al., 2018). Inherently, these techniques are data inefficient due to the complexity of the problem. One way to address this issue is to use transfer learning, such as transfer from pre-trained policies (Rusu et al., 2015). However, a teacher policy is not always available for the main task. When in this scenario, one can train policies in other tasks that share enough similarities and hope for a positive transfer. One way of exploiting

this extra information is to use behavioral cloning, or distillation (Hinton et al., 2015; Rusu et al., 2015), to guide the main task in its initial learning phase (Schmitt et al., 2018), although it might be difficult to find a suitable strategy that combines the main and auxiliary losses and/or smoothly transition between them. Typically, the teacher policy can be treated as an auxiliary loss (Schmitt et al., 2018) or a prior (Teh et al., 2017) with a fixed mixing coefficient. However, these techniques become unsound if the teacher policy is helpful only in specific states while hindering in other states.

We propose a simple RL experiment to show that our method is capable of finding the strategy of combining the main loss and the auxiliary loss. We define a distribution over a set of $15 \times 15$ gridworlds, where an agent observes its surrounding (up to four pixels away) and can move in four directions.We randomly place two types of positive rewards, $+5$ and $+10$ points, both terminating an episode. In order to guarantee a finite length of episodes, we add a fixed probability of $0.01$ of transitioning to a non-rewarding terminal state. Experiment details are provided in Appendix F.

First, we train a Q-learning agent on such a gridworld which gives us a *teacher* policy $\pi^Q$. Then, we create the main task to which there is a possible positive knowledge transfer by keeping the environment with the same gridworld layout but remove the $+10$ rewards (and corresponding states are no longer terminating). Consequently, we have two tasks: the auxiliary task $\mathcal{T}_{aux}$ where we have a strong *teacher* policy $\pi^Q$, and the main task $\mathcal{T}_{main}$ where the $+10$ rewards are removed. We sample $1,000$ such environment pairs and report expected returns obtained (100 evaluation episodes per evaluation point) using various training regimes. One can use any RL method to solve the main task and learn $\pi$, here we use episode-level policy gradient (Williams, 1992) with value function as a baseline method, which gives a score of slightly above **1** point after $10,000$ steps of training (see the top row of Figure 3).

To leverage teacher policies, we define the auxiliary loss to be a distillation loss, which is a per-state cross-entropy between teacher's and student's distributions over actions. First, we test using solely the distillation loss while sampling trajectories from the student. We recover a subset of teacher's behaviors and end up with **0** point—an expected negative transfer as the teacher is guiding us to states that are no longer rewarding. Then, we test simply adding gradients estimated by policy gradient and distillation. The resulting policy learns quickly but saturates at a return of **1** point, showing very limited positive transfer. Lastly, when using our proposed gradient cosine similarity as the measure of transferability, we get a significant performance boost. The learned policies reach baseline performance after just one-third of steps taken by the baseline, and on average obtain **3** points after $10,000$ steps.[2] See Figure 3 for all learning curves. In Appendix F, we visualize the environment and show an example solution.

This experiment shows that gradient cosine similarity allows using knowledge from other related tasks in an automatic fashion. The agent is simply ignoring teacher signal when it disagrees with policy gradient estimator. If they do agree in terms of which actions to reinforce—teacher logits are used for better replication of useful policies. In particular, in the bottom row of Figure 3, we present an experiment of transfer between the same task $\mathcal{T}_{main}$. We see that the cosine similarity experiments underperformed that of simply adding the two losses. This is expected as the noise in the gradients makes it hard to measure if the two tasks are a good fit or not.

## 3.3 Experiments on Atari

Finally, we consider a similar RL setup on the Atari domain (Bellemare et al., 2013). For this set of experiments, we follow the same convolutional architecture as in previous works (Mnih et al., 2015; 2016; Espeholt et al., 2018; Hessel et al., 2018) and train using the batched actor-critic with V-trace algorithm (Espeholt et al., 2018). Details on the experiment setup are provided in Appendix G.

First, we look at training an agent to play a main task (here, Breakout) given a *sub-optimal* teacher solution to the task. Analogous to the previous experiment, we leverage information about the task by distilling the teacher's behaviour with a Kullback-Leibler (KL) loss. As expected, solely relying on distilling from the sub-optimal teacher (*Only KL*) leads to lower performance. Training with both distillation and RL losses (*RL+KL(Baseline)*) leads to slightly better but also sub-optimal performance. While both approaches learn very quickly, they plateau much lower than the pure RL

---

[2]Note that we compute cosine similarity between a distillation gradient and a single sample of the policy gradient estimator, meaning that we are using a high variance estimate of the similarity. For larger experiments, one would need to compute running means for reliable statistics.

**Cross-task transfer experiment $\mathcal{T}_{aux} \to \mathcal{T}_{main}$**

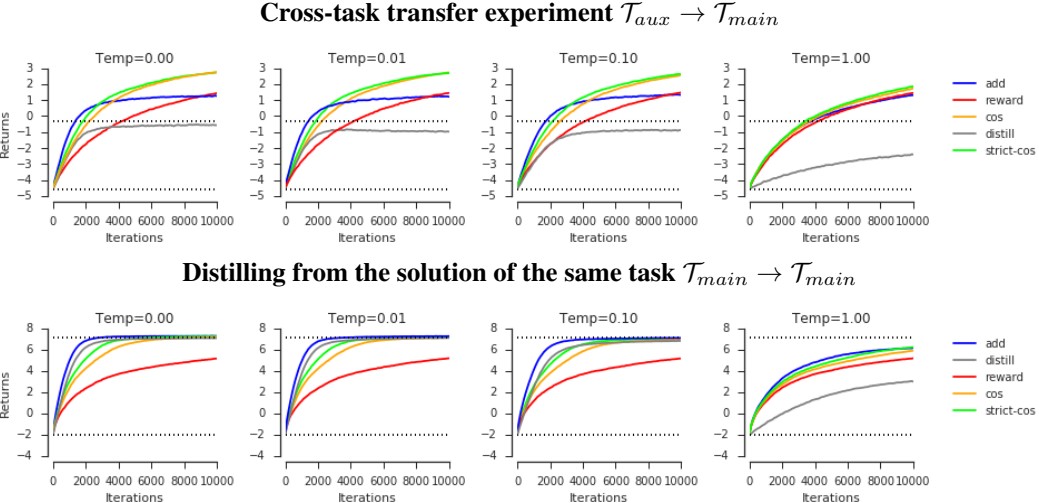

**Distilling from the solution of the same task $\mathcal{T}_{main} \to \mathcal{T}_{main}$**

Figure 3: *Top row:* expected learning curves for cross-environment distillation experiments, averaged over $1,000$ partially observable gridworlds. Teacher's policy is based on Q-Learning, its performance in a new environment (with modified positive rewards) is represented by the top dotted line. The bottom dotted line represents random policy score. Each column represents a different temperature applied to the teacher policy. 0 temperature refers to the original deterministic greedy policy given by Q-Learning. We report five methods: **reward** using just policy gradient in the new task; **distill** using just distillation cost towards the teacher; **add** adding the two above; **cos** using the weighted version of our method (Algorithm 2); **strict cos** using the unweighted version of our method (Algorithm 1). *Bottom row:* expected learning curves for same-environment distillation experiments when the teacher is perfect. In this case, the optimal thing is to trust the teacher everywhere.

approach (*RL(Baseline)*). In our method (*RL+KL(Our Method)*), the KL penalty is scaled at every time-step by the cosine similarity between the policy gradient and distillation losses; once this falls below a fixed *threshold*, the loss is 'turned off'. Figure 4 shows that our approach is able to learn quickly at the start but continue fine-tuning with pure RL loss once the distillation loss is zeroed out.

Lastly, we consider a setting where the main task $\mathcal{T}_{main}$ is to train an agent to play two Atari games, Breakout and Ms. PacMan. Similar to previous experiment, we have access to a teacher trained on just Breakout as the auxiliary task $\mathcal{T}_{aux}$, from which we distill a policy via KL loss. Note that $\mathcal{T}_{main}$ itself is chosen to be *Multitask* to illustrate a complex scenario where $\mathcal{T}_{aux}$ helps with only part of $\mathcal{T}_{main}$, and that too only initially. We consider a distillation loss as was done previously by adding the auxiliary KL loss $\mathcal{L}_{aux}$ (between the teacher and student policies) to the RL multi-task loss $\mathcal{L}_{main}$ at every time step. Intuitively, doing so would result in the agent only be able to solve one of the tasks—the one the teacher knows about, as the gradients from distillation loss would interfere with the policy gradient. Figure 5 shows that, compared to the baseline *Multitask* and the simple addition of *Multitask RL + Distillation* approaches where the agent learns one task at the expense of the other, our method of scaling the auxiliary loss by gradient cosine

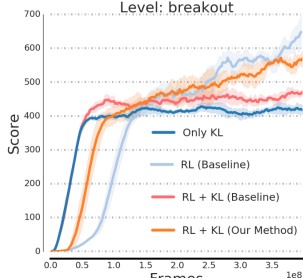

Figure 4: Results on Breakout. We look at the effects of distilling a sub-optimal policy as an auxiliary task.

similarity is able to compensate for this by learning from the teacher and then turning off the auxiliary distillation; it learns Ms. PacMan without forgetting Breakout. The evolution of the gradient cosine similarity between Breakout and Ms. PacMan provides a meaningful cue for the usefulness of $\mathcal{L}_{aux}$.

## 4   RELATED WORK

Our work is related to the literature on identifying task similarity in transfer learning. It is generally believed that *positive transfer* can be achieved when *source* task(s) and *target* task(s) are related.

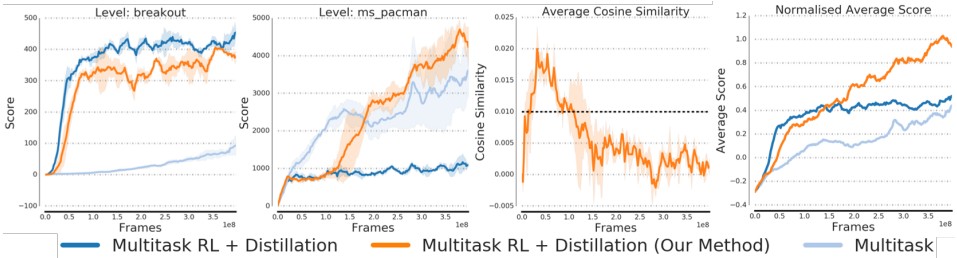

Figure 5: Results on Breakout and Ms. PacMan (averaged over 3 seeds). The two plots to the left show performance on Breakout and Ms. PacMan respectively. The third plot shows how the gradient cosine similarity between the two tasks changes during training. The last plot shows an average score of the multi-task performance (normalized independently for each game based on the best score achieved across all experiments). Our method is able to learn both games without forgetting and achieves the best average performance.

However, it is usually assumed that this relatedness mapping is provided by human experts (Taylor and Stone, 2009; Pan et al., 2010); few works have addressed the problem of finding a general measure of similarity to predict transferability between tasks. In image classification, Yosinski et al. (2014) defined image similarity in ImageNet by manually splitting classes into man-made versus natural objects. In RL, methods have been proposed to use the Markov decision process (MDP) similarity as a measure of task relatedness (Carroll and Seppi, 2005; Ammar et al., 2014). While in some degree capture task similarity, these measures are often domain-specific and not generalizable. In addition, none of these works have explicitly used the learned similarity metric to improve performance. In our work, we propose to use cosine similarity of gradients as a generalizable measure across domains and show it can be directly leveraged to improve the performance of the main task. One important aspect of task similarity for transfer is that it is highly dependent on the parametrization of the model and current value of the parameters. We exploit this property by providing a heuristic similarity measure for the current parameters, resulting in an approach that relies on an adaptive weight over the updates of the model.

Auxiliary tasks have shown to be beneficial in facilitating learning across domains. In image classification, Zhang et al. (2016) used unsupervised reconstruction tasks. In RL, the UNREAL framework (Jaderberg et al., 2017) incorporates unsupervised control tasks along with reward prediction learning as auxiliary tasks. Mirowski et al. (2017) studied auxiliary tasks in the context of navigation. Papoudakis et al. (2018) also explored auxiliary loses for VizDoom. However, these works rely on empirical results and do not address how the auxiliary tasks were selected in the first instance. In this work, we aim to propose a simple yet effective way of explicitly selecting auxiliary tasks by using cosine similarity of task gradients.

Our work is also related to multi-task learning (Caruana, 1997), particularly the line of work on using adaptive scaling techniques for multi-objective learning. For example, a recently developed algorithm, *GradNorm* (Chen et al., 2018), uses gradient magnitude to scale loss function for each task, aiming to learn well for all tasks. Similarly, Kendall et al. (2018) proposed a weighting mechanism by considering the homoscedastic uncertainty of each task. However, our work is different in two ways: first, in our problem setup, we care only about the performance of the main task and we do not care about all tasks; hence, the optimization goal is different from their setup which is more similar to traditional multi-objective optimization. Furthermore, it is important to note that our method differs from aforementioned work in that they scale the losses individually without looking at their interaction (which can lead to poor performance in our problem setup, when the auxiliary task hurts the main task), whereas we look for alignments in the vector field between the main and the auxiliary task, and the auxiliary task is used only when it is well-aligned with the main task.

## 5    DISCUSSION

In this work, we explored a simple yet efficient technique to ensure that an auxiliary loss does not hurt the learning on the main task. The proposed approach reduces to applying gradients of the auxiliary task only if they are a descent direction of the main task.

We discuss here a few shortcomings of this method. First, estimating the cosine similarity between the gradients of tasks could be expensive or noisy and that the threshold for turning off the auxiliary is a fixed constant. These could be addressed by calculating a running average of the cosine similarity to get a smoother result and potentially hyper-tune the threshold instead of setting it as a fixed constant. One might argue that our approach would fail in high-dimensional spaces since random vectors in such spaces tend to be orthogonal, so that cosine similarity will be naturally driven to 0. In fact, this is not the case; if two gradients are meant to be co-linear, the noise components cancel each other thus will not affect the cosine similarity estimation. We empirically explore this in Appendix H. Second, the new loss surface might be less smooth which can be problematic when using optimizers that rely on statistics of the gradients or second order information (e.g. Adam or RMSprop). In these cases, the transition from just the gradient of the main task to the sum of the gradients can affect the statistics of the optimizer in unwanted ways.

Lastly, although the proposed approach works well empirically on complex and noisy tasks like Atari games, as discussed in Section 2, it guarantees only that the main task will converge, but not how fast it will be. While removing the worst case scenarios is important and a good first step, one might care more for data efficiency when using auxiliary losses (i.e., faster convergence). Particularly in Appendix D Figure 8, we construct a counter-example where the proposed update rule slows down learning, compared to optimizing the main task alone. Nevertheless, we have empirically shown the potential of using the proposed hypothesis as a simple yet efficient way of picking a suitable auxiliary task. While we have mostly considered scenarios where the auxiliary task helps initially but hurts later, it would be interesting to explore settings where the auxiliary task hurts initially but helps in the end. Examples of such are annealing $\beta$ in $\beta$-VAE (Higgins et al., 2017) and annealing the confidence penalty in (Pereyra et al., 2017).

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

## A    PROOFS

### A.1    PROOF FOR PROPOSITION 1

*Given any gradient vector field $G(\boldsymbol{\theta}) = \nabla_{\boldsymbol{\theta}}\mathcal{L}(\boldsymbol{\theta})$ and any vector field $V(\boldsymbol{\theta})$ (such as gradient of another loss function, but could be arbitrary set of updates), an update rule of the form*

$$\boldsymbol{\theta}^{(t+1)} := \boldsymbol{\theta}^{(t)} - \alpha^{(t)}(G(\boldsymbol{\theta}^{(t)}) + V(\boldsymbol{\theta}^{(t)})\max(0, \cos(G(\boldsymbol{\theta}^{(t)}), V(\boldsymbol{\theta}^{(t)}))))$$

*converges to the local minimum of $\mathcal{L}$ given small enough $\alpha^{(t)}$.*

*Proof.* Let us denote

$$G^{(t)} := G(\boldsymbol{\theta}^{(t)}) \qquad V^{(t)} := V(\boldsymbol{\theta}^{(t)}) \qquad \nabla\mathcal{L}^{(t)} := \nabla_{\boldsymbol{\theta}}\mathcal{L}(\boldsymbol{\theta}^{(t)})$$

$$\Delta\boldsymbol{\theta}^{(t)} := G^{(t)} + V^{(t)}\max(0, \cos(G^{(t)}, V^{(t)})).$$

Our update rule is simply $\boldsymbol{\theta}^{(t+1)} := \boldsymbol{\theta}^{(t)} - \alpha^{(t)}\Delta\boldsymbol{\theta}^{(t)}$ and we have

$$\langle\Delta\boldsymbol{\theta}^{(t)}, \nabla\mathcal{L}^{(t)}\rangle = \langle G^{(t)} + V^{(t)}\max(0, \cos(G^{(t)}, V^{(t)})), \nabla\mathcal{L}^{(t)}\rangle \tag{3}$$

$$= \langle G^{(t)}, \nabla\mathcal{L}^{(t)}\rangle + \langle V^{(t)}\max(0, \cos(G^{(t)}, V^{(t)})), \nabla\mathcal{L}^{(t)}\rangle \tag{4}$$

$$= \|\nabla\mathcal{L}^{(t)}\|^2 + \frac{1}{\|V^{(t)}\|\|\nabla\mathcal{L}^{(t)}\|}\max(0, \langle\nabla\mathcal{L}^{(t)}, V^{(t)}\rangle)\langle V^{(t)}, \nabla\mathcal{L}^{(t)}\rangle \geq 0. \tag{5}$$

And it can be 0 if and only if $\|\nabla\mathcal{L}^{(t)}\| = 0$ (since sum of two non-negative terms is zero iff both are zero, and step from (4) to (5) is only possible if this is not true), thus it is 0 only when we are at the critical point of $\mathcal{L}$. Thus the method converges due to convergence of steepest descent methods, see "Cauchy's method of minimization" (Goldstein, 1962). ☐

### A.2    PROOF FOR PROPOSITION 3

*In general, the proposed update rule does not have to create a conservative vector field.*

*Proof.* Proof comes from a counterexample, let us define in 2D space:

$$\mathcal{L}_{main}(\theta_1, \theta_2) = a\theta_1$$

$$\mathcal{L}_{aux}(\theta_1, \theta_2) = \begin{cases} a\theta_1 & \text{if } \theta_1 \in [1, 2] \wedge \theta_2 \in [0, 1] \\ 0 & \text{therwise} \end{cases}$$

for some fixed $a \neq 0$. Let us now define two paths (parametrized by $s$) between points $(0, 0)$ and $(2, 2)$, path $A$ which is a concatenation of a line from $(0, 0)$ to $(0, 2)$ (we call it $U$, since it goes up) and line from $(0, 2)$ to $(2, 2)$ (which we call $R$ as it goes right), and path $B$ which first goes right and then up. Let $V_{\cos}$ denote the update rule we follow, then:

$$\int_A V_{\cos}ds = \int_A \nabla\mathcal{L}_{main}ds = \int_U \nabla\mathcal{L}_{main}ds + \int_R \nabla\mathcal{L}_{main}ds = \int_R \nabla\mathcal{L}_{main}ds = 2a$$

At the same time, since gradient of $\mathcal{L}_{main}$ is conservative by definition:

$$\int_B V_{\cos}ds = \int_B \nabla\mathcal{L}_{main}ds + \int_C \nabla\mathcal{L}_{aux}ds = \int_A \nabla\mathcal{L}_{main}ds + \int_C \nabla\mathcal{L}_{aux}ds = 2a + \int_C \nabla\mathcal{L}_{aux}ds = 3a$$

where $C$ is a part of $B$ that goes through $[1, 2] \times [0, 1]$. We conclude that $\int_A V_{\cos}ds \neq \int_B V_{\cos}ds$, so our vector field is not path invariant, thus by Green's Theorem it is not conservative, which concludes the proof. See Figure 6 for visualization. ☐

## B    ONE-DIMENSIONAL TOY EXAMPLE

Figure 7 shows the surfaces along with gradients for the one-dimensional motivating example described in Section 2.

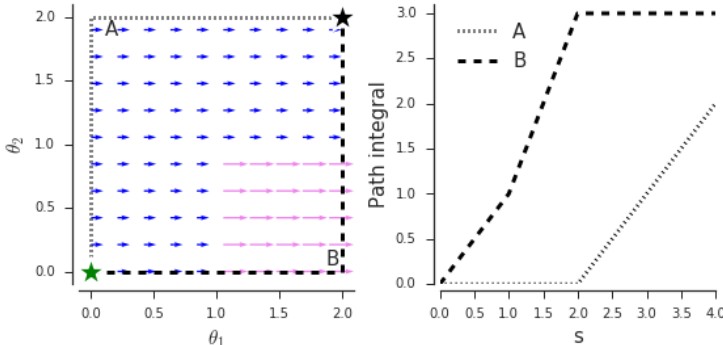

Figure 6: Visualization of the counterexample from Proposition 3, stars denote starting (green) and end (black) points. Dotted and dashed lines correspond to paths A and B respectively. Blue arrows represent gradient vector field of the main loss, while the violet ones the merged vector field.

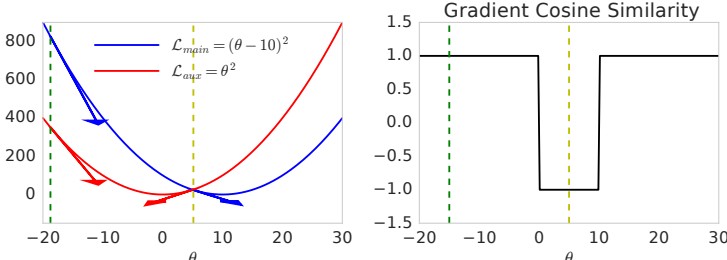

Figure 7: Illustration of cosine similarity between gradients on synthetic loss surfaces.

## C WEIGHTED VERSION OF OUR METHOD

Algorithm 2 describes the *weighted* version of our method.

## D TOY EXAMPLE SHOWING SLOW-DOWN

We discuss here a few potential issues of using cosine similarity of gradients to measure task similarity. First, the method depends on being able to compute cosine between gradients. However, in DL we rarely are able to compute exact gradients in practice, nut instead depend on their high variance estimators (mini batches in supervised learning, or Monte Carlo estimators in RL). Consequently, estimating the cosine similarity might require additional tricks such as keeping moving averages of estimates. Second, adding additional task gradient in selected subset of iterates can lead to very bumpy surface from the perspective of optimizer, causing methods which keep track of gradient statistics/estimate higher order derivatives, can be less efficient. Finally, one can construct specific functions, where despite still minimizing the loss, one significantly slows down optimization process. Figure 8 provides one such function as an example.

## E IDENTIFYING NEAR AND FAR CLASSES IN IMAGENET

As a ground truth for class similarity, we identify pairs of ImageNet classes to be near or far using, *lowest common ancestor (LCA)* and *Frechet Inception Distance (FID)* (Heusel et al., 2017).

ImageNet follows a tree hierarchy where each class is a leaf node. We define the distance between a pair of classes as at which tree level their LCA is found. In particular, there are 19 levels in the class tree, each leaf node (i.e. class) is considered to be level 0 while the root node is considered to be level 19. We perform bottom-up search for one pair of random sampled classes and find their LCA node—the class distance is then defined as the level number of this node. For example, class 871 ("trimaran") and class 484 ("catamaran") has class distance 1 because their LCA is one level up.

---

**Algorithm 2** Weighted version of our method.

1: Initialize shared parameters $\theta$ and task specific parameters $\phi_{main}, \phi_{aux}$. randomly.
2: **for** iter $= 1 :$ max_iter **do**
3:      Compute $\nabla_{\theta}\mathcal{L}_{main}, \nabla_{\phi_{main}}\mathcal{L}_{main}, \nabla_{\theta}\mathcal{L}_{aux}, \nabla_{\phi_{aux}}\mathcal{L}_{aux}$.
4:      Update $\phi_{main}$ and $\phi_{aux}$ using corresponding gradients
5:      Update $\theta$ using $\nabla_{\theta}\mathcal{L}_{main} + \max(0, \cos(\nabla_{\theta}\mathcal{L}_{main}, \nabla_{\theta}\mathcal{L}_{aux}))\nabla_{\theta}\mathcal{L}_{aux}$

---

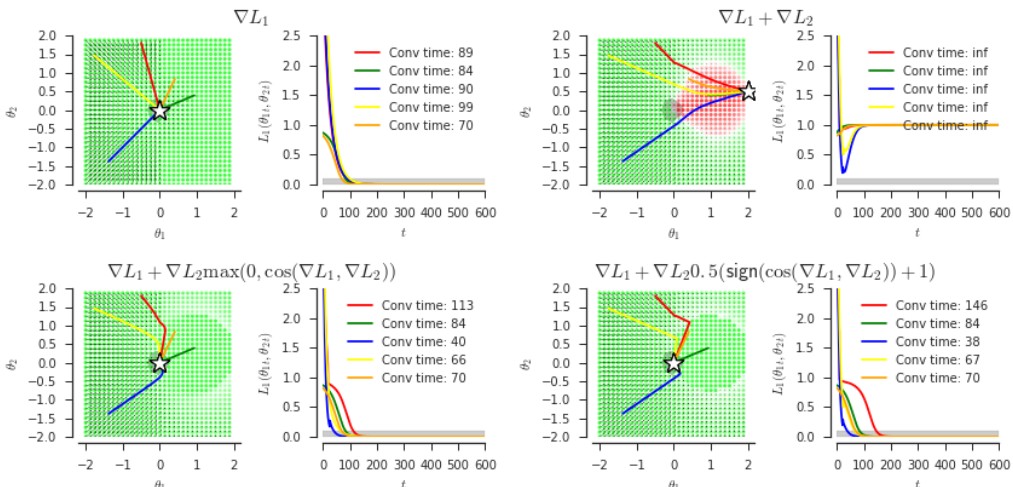

Figure 8: Negative example optimization for $L_1(\theta) = (\theta_1 < 0)(\theta_1^2 + \theta_2^2) + (\theta_1 > 0)\Big(1 - \exp\big(-2(\theta_1^2 + \theta_2^2)\big)\Big)$ and $L_2(\theta) = (\theta_1 - 2)^2 + (\theta_2 - 0.5)^2$ where the proposed method slows down the process (compared on red runs). For the ease of presentation, we choose $L_1$, which is non-differentiable/smooth when $\theta = 0$. But one can create any smooth functions with analogous properties. The core idea is, when there exists a flat region on the loss surface, the auxiliary lost tends to push the iterates to this region. Even though this move still decreases the loss (i.e., convergence is guaranteed), the optimization process will be slowed down.

FID is used as a second measure of similarity. We obtain the image embedding of a pair of classes using the penultimate layer of a pre-trained ResNetV2-50 model (He et al., 2016b) and then compute the embedding distance using FID, defined in Heusel et al. (2017) as:

$$\text{FID} = d^2\big((m_1, C_1), (m_2, C_2)\big) = \|m_1 - m_2\|_2^2 + \text{Tr}\big(C_1 + C_2 - 2(C_1 C_2)^{1/2}\big). \qquad (6)$$

where $m_k, C_k$ denote the mean and covariance of the embeddings from class $k$.

We randomly sampled 50 pairs of classes for each level of $LCA = \{1, 2, 3, 4, 16, 17, 18, 19\}$ (400 pair of classes in total) and compute their FID. Figure 9 shows a plot of LCA (x-axis) verses FID (y-axis) over our sampled class pairs. It can be seen that LCA and FID are (loosely) correlated and that they reflect human intuition of task similarity for some pairs. For example, *trimaran and catamaran* (bottom-left) are similar both visually and conceptually, whereas *rock python and traffic light* (top-right) are dissimilar both visually and conceptually. However, there are contrary examples where LCA disagrees with FID; *monkey pinscher and doberman pinscher* (top-left) are visually dissimilar but conceptually similar, whereas *bubble and sundial* (bottom-right) are visually similar but conceptually dissimilar.

Per the observations, in subsequent experiments we pick class pairs that are {*Low LCA, Low FID*} as *near* pairs (e.g., trimaran and catamaran), and class pairs that are {*high LCA, high FID*} as *far* pairs (e.g., rock python and traffic light).

## F   GRIDWORLD EXPERIMENTS

We define a distribution over $15 \times 15$ gridworlds, where an agent observes its surrounding (up to 4 pixels away) and can move in 4 directions (with 10% transition noise). We randomly place walls

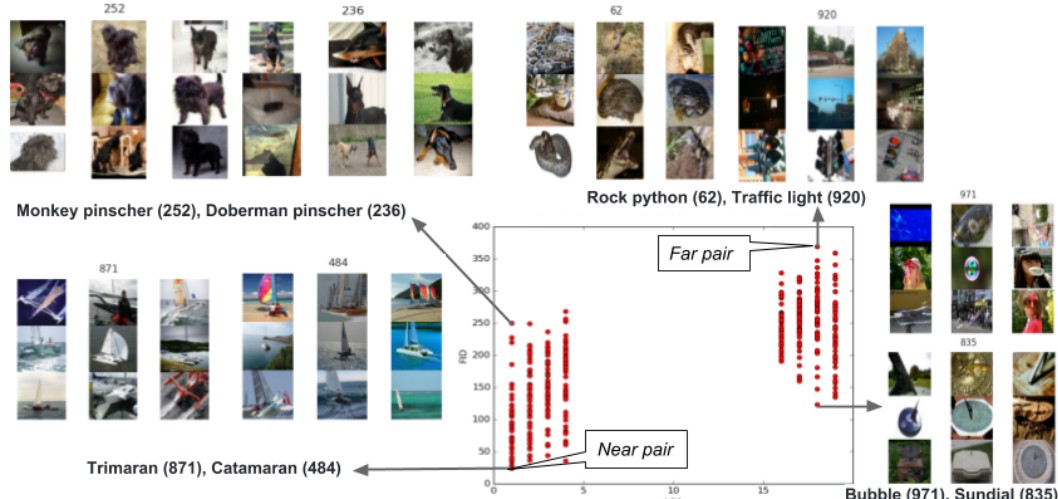

Figure 9: LCA ($x$-axis) versus FID ($y$-axis) as a ground truth for class similarity. The measurements reflect human intuition of class similarity; *trimaran and catamaran* (bottom-left) are similar both visually and conceptually, whereas *rock python and traffic light* (top-right) are dissimilar both visually and conceptually.

(blocking movement) as well as two types of positive rewards: $+5$ and $+10$ points, both terminating an episode. There are also some negative rewards (both terminating and non-terminating) to make problem harder. In order to guarantee (expected) finite length of episodes we add fixed probability of $0.01$ of transitioning to a non-rewarding terminal state.

For the sake of simplicity we use episode-level policy gradient (Williams, 1992) with value function baseline, with policies parametrized as logits $\boldsymbol{\theta}$ of $\pi(a|s) = \frac{\exp(\theta_{s,a})}{\sum_b \exp(\theta_{s,b})} \in [0,1]$, baselines as $B_s \in \mathbb{R}$, with fixed learning rate of $\alpha = 0.01$, discount factor $\gamma = 0.95$ and 10,000 training steps (states visited).

For this setup, the update rule for each sequence $\tau = \big((s_1, a_1, r_1), \dots (s_N, a_N, r_N)\big)$ is thus given by

$$\Delta\boldsymbol{\theta} = \alpha \nabla_{\boldsymbol{\theta}} \log \pi(a_{t'}|s_{t'}) \left[ \sum_{i=0}^{N-t'} r_{t'+i} - B_{s_{t'}} \right] = \alpha G^{(t)} \qquad \Delta B_{s_{t'}} = -\alpha \nabla_{B_{s_{t'}}} \big(B_{s_{t'}} - \sum_{i=0}^{N-t'} r_{t'+i}\big)^2 .$$

In order to make use of expert policies for $\mathcal{T}_{aux}$ we define auxiliary loss as a distillation loss, which is just a per-state cross-entropy between teacher's and student's distributions over actions. If we just add gradients estimated by policy gradient, and the ones given by distillation, the update is given by

$$\Delta\boldsymbol{\theta} = \alpha \left[ G^{(t)} - \nabla_{\boldsymbol{\theta}} H^{\times}(\pi^Q(\cdot|s_{t'}) \| \pi(\cdot|s_{t'})) \right] = \alpha [ G^{(t)} + \sum_a \pi^Q(a|s_{t'}) \nabla_{\boldsymbol{\theta}} \log \pi(a|s_{t'}) ],$$

where $V^{(t)} = \sum_a \pi^Q(a|s_{t'}) \nabla_{\boldsymbol{\theta}} \log \pi(a|s_{t'})$ and $H^{\times}(p,q) = -\sum_k p_k \log q_k$ is the cross entropy.

However, if we use the proposed gradient cosine similarity, we get the following update

$$\Delta\boldsymbol{\theta} = \alpha \left[ G^{(t)} + V^{(t)} \big( 2 \cdot \mathrm{sign}(\cos(G^{(t)}, V^{(t)})) - 1 \big) \right].$$

This get a significant boost to performance, and policies that score on average **3** points after 10,000 steps and obtain baseline performance after just one third of steps. Figure 10 shows a depiction of the task and an example solution.

## G    ATARI EXPERIMENTS

For these experiments, we use a convolutional architecture as in previous work (Espeholt et al., 2018; Hessel et al., 2018; Mnih et al., 2015; 2016), trained with batched actor-critic with the V-trace

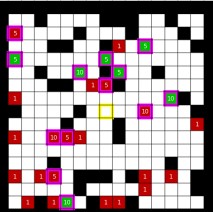 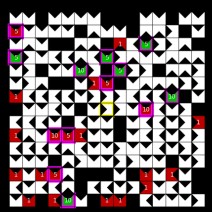 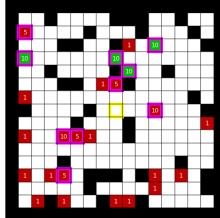 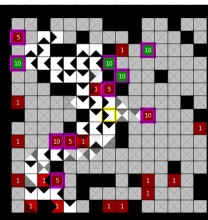

Figure 10: **Left most**: Initial task $\mathcal{T}_{main}$, yellow border denotes starting point and violet ones terminating states. Red states are penalizing with the value in the box while the green ones provide positive reward. **Middle Left**: Solution found by a single run of Q-learning with uniform exploration policy. **Middle Right**: Transformed task $\mathcal{T}_{aux}$. **Right most**: Solution found by gradient cosine similarity driven distillation with policy gradient.

algorithm (Espeholt et al., 2018). We use a learning rate of $0.0006$ and an entropy cost of $0.01$ for all experiments, with a batch size of $32$ and $200$ parallel actors.

For the single game experiment, Breakout, , we use $0.02$ for the threshold on the cosine similarity and, for technical reasons we ended up computing the cosine distance on a per-layer basis and then averaged. We additionally need to do a moving average of the cosine over time $(0.999c^{(t-1)} + 0.001c^{(t)})$ to ensure there are no sudden spikes in the weighting due to noisy gradients. Same setting is used for the multi-task experiment, just that the threshold is set to $0.01$.

## H   COSINE SIMILARITY IN HIGH DIMENSIONS

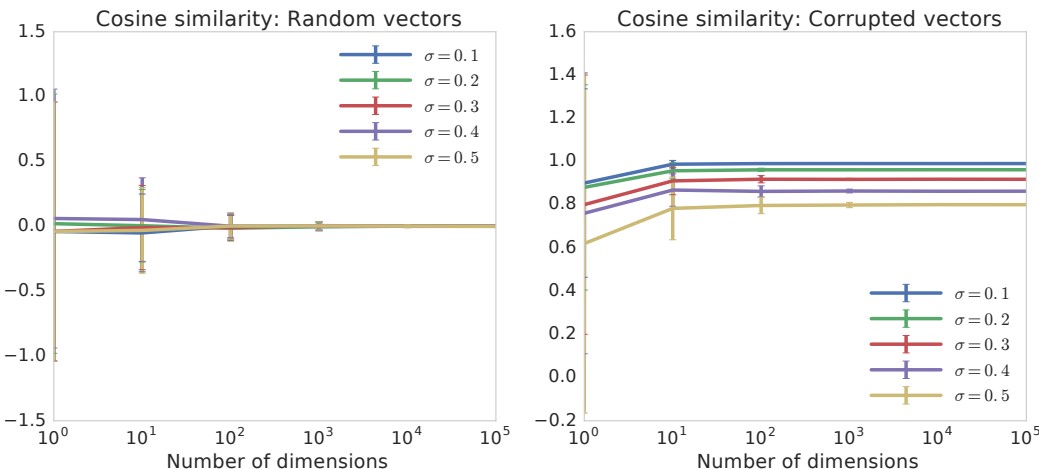

Figure 11: Cosine similarity as a function of dimensionality. On the left, we generate two random vectors $\theta_1$ and $\theta_2$ from a Gaussian distribution with zero mean and variance $\sigma^2$ and as expected, the cosine similarity drops to zero very quickly as the number of dimensions increases. On the right, we mimic a scenario where the true gradients of the main and auxiliary are aligned, however we observe only *corrupted* noisy gradients which are noisy copies of the true underlying vector; we generate $\mu \sim \mathcal{N}(0, I_d)$ and generate $\theta_1 \sim \mathcal{N}(\mu, \sigma I_d)$ and $\theta_2 \sim \mathcal{N}(\mu, \sigma I_d)$. In this case, cosine similarity is larger in higher dimensions (as the inner product of the corruption noise goes to zero).

