# OpenReview forum: "Adapting Auxiliary Losses Using Gradient Similarity"
_ICLR.cc/2019/Conference_

### Official Review · AnonReviewer2 · 2018-10-31
**Simple method for using gradient information of auxiliary task when it agrees with gradient of main task**

**Rating:** 6
**Confidence:** 3

**Review:**

The paper proposes a method for using auxiliary tasks to support the optimization with respect to a main task. In particular, the method assumes the existence of a loss function for the main task that we are interested in, and a loss function for an auxiliary task that shares at least some of the parameters with the main loss function. When optimizing for the main loss function, the gradient of the auxiliary loss function is also used to update the shared parameters in cases of high cosine similarity with the main task. The method is demonstrated on image classification and a few reinforcement learning settings.

The idea of the paper is simple, and the method has a nice property of (if ignoring some caveats) guaranteeing steps that are directionally correct with respect to the main task. In that sense it is useful in practice, as it limits the potential damage the auxiliary task does to the optimization of the main task.

As the authors also note, the method suffers from some drawbacks. Although the method limits the negative effect of the auxiliary task on the optimization of the main loss function, it can still slow down optimization if the auxiliary task is not well chosen. In that sense, the method is no silver bullet. In addition, the method seems fairly computationally expensive (it would be interesting to understand how much it slows down an update, I would assume the added complexity is roughly a constant multiplier). However, as an alternative to naively adding an auxiliary task, the proposed method is a welcome addition to the tool box of practitioners.

Although the experiments presented in the paper are quite different from each other, I would have wished for even more experiments. The reason is that as the method does not guarantee faster convergence, its applicability is mainly an empirical question. Especially experiments where auxiliary tasks have been used before would be interesting to test with the only addition being introducing the method proposed.

The paper is generally well written and the results are fairly clearly presented. As a minor comment, the authors might want to check that articles (such as "the") are not missing in the text.

All in all, the main merit of the proposed method is its conceptual simplicity and easy to understand value in practical applications where an auxiliary loss function is available. The method also seems to work well enough in the experiments presented.

---

> ### Author Response · Authors · 2018-11-27
> **Response to Reviewer 2**
>
> We thank Reviewer 2 for the feedback!
>
> Regarding the reviewer’s concern on computation expenses of our method, our method does not add expensive computation because the only additional step is to compute a one-step cosine similarity between gradients. Since computing cosine similarity is of O(D) time complexity where D is the dimensionality of the shared parameters, it is just a constant multiplier, as already noted by the reviewer.
>
> Regarding the reviewer’s suggestion on running more experiments: we believe that our experiments support our main claims and as the reviewer noted, we already report a diverse set of experiments. For the suggestion of “experiments where auxiliary tasks have been used before would be interesting to test with the only addition being introducing the method proposed.”, note that this may not be necessarily the best setup to illustrate our method as some of the prior work could have selected tasks such that the auxiliary task always helps (i.e. the cosine similarity is always greater than 0). Also the problem setup and motivations are slightly different in some of the prior work. For example, in Mirowski et al., the use of depth prediction task as the auxiliary task was meant to assist “representation learning”, which is different from the goal of our method that the auxiliary task seeks “alignment” with the main task in their gradient space (e.g., in the RL GridWorld task, our setting has no representation learning at all since policies are tabular).
>
> We believe that this method is a useful addition to the tool box (as the reviewer also observed) and opens up the possibilities of using auxiliary tasks which may help initially but hurt in the end. We hope that the simplicity of our method will encourage the community to try these on other domains and investigate extensions that further improve performance.
>
> Thanks for pointing out the writing mistakes. We have fixed them in the revised version.

---

### Official Review · AnonReviewer1 · 2018-11-01
**Used grandent similarity to decide whether an auxiliary task is useful or hurting the main task. Showed improved results in supervised learning and reinforcement learning domains.**

**Rating:** 6
**Confidence:** 4

**Review:**

The paper studies the problem of how to measure the similarity between an auxiliary task and the target tasks, and further decide when to use the auxiliary loss in the training epoches. The proposed cosine simiarity based soft gradient update scheme seems reasonable. The author(s) also experiment the proposed method on three tasks, one supervised learning image classification task, two reinforcement learning tasks, and show improved results respectively.

The paper is in generally well-written. However it would be great if the concerns below could be addressed or discussed in the paper.

1) The proposed method is based on the intuition: if the gradients of the target and auxiliary loss are in the same direction, the auxiliary loss will help the main/target task. Some examples are showed in the paper to support this argument, however it would be helpful if there is some theoritical gurantee on this. So a more general question would be: rather than define the similarity measure to measure the gradient similarity of the target and auxiliary loss, it would be more useful to try to learn or define whether the auxiliary task is good for the target task beforehand.

2) In proposition 1, if the concerns in 1) are reasonable, the equation would be doubtful. For example, one can simply try (g(target task)-g(auxiliary task)) in the equation. Besides, more similarity metrics are expected to be compared here to show why cosine is the optimal choice. For example, L2.

3) Too much content is embedded in appendix, for example, it would be helpful to move the two algorithms or at least discussed the two variants of the gradient updates in the experimental section. Since it is not clear to me whether hard cosine mixing or soft cosine mixing is used to produce the results in the image classification task.

4) In the image classification task, a quantitative analysis would be more convincing since the semantics of the near and far is really hard to define. Even the authors can show a vague definition, it will be helpful. In figure 2b), why the cosine method performs worse compared the other methods before 5000 in x-axis? Is this because of the noise of the gradient? Plus, what is the optimizer used in this experiment?

5) In the first reinforcement learning task, since cosine similarity is the only method used to measure the similarity between auxiliary task and the target task, it would be useful to show the comparison among other task relatedness method in reinforcement learning. For 'This is expected as the noise in the gradients make it hard to measure if the two tasks are a good fit or not',  why is this? Since cosine similarity would be zero if the two tasks are not good fit.

---

> ### Author Response · Authors · 2018-11-27
> **Response to Reviewer 1**
>
> We thank Reviewer 1 for the feedback! We have revised the paper based on your suggestions and we also address your concerns individually below.
>
> In the first review point, the reviewer asked for a theoretical guarantee of our method. We have shown in section 2 that our method has a convergence guarantee (although there is no guarantee of an improvement of the speed of convergence). Proof details were presented in Appendix A.1. We agree that defining the usefulness of an auxiliary task beforehand is an interesting future direction. We believe that the level of usefulness has to be conditioned on the model (or at least a family of models) and potentially on the initialization of parameters. In some regard, our proposed approach relies on conditioning on all of this information (e.g. the value of theta) and it would be interesting if some of this can be efficiently marginalized away.
>
> For the second review point, it is unclear to us that another metric (and definitely not true for L2) can guarantee convergence on the task of interest. Based on our theoretical proof in section 2, cosine similarity is the metric for which we can prove the convergence and hence the only one that is applicable for our approach. In particular, we believe that instead of thinking of cosine as a metric, a fruitful perspective is to see it as a projection of the gradient of the auxiliary loss in the subspace of decent directions of the main loss.
>
> For the third review point, we have revised our paper and included Algorithm 1 in the main text. We will move some more content from appendix to the main text for the final camera ready.
>
> For the fourth review point, we did not identify the near or far class pairs in ImageNet based on the semantic similarity of class names. Instead, we used two semantic quantitative measures, the lowest common ancestor (LCA) in the tree hierarchy and the Frechet Inception Distance (FID) of pre-trained image embeddings. The procedure is described briefly in section 3.1 and detailed in Appendix E. We believe they have quantitatively defined a good similarity metric for our setting. In figure 2b, our cosine method did not perform worse before step 5000. In fact, the cosine method has shown a jumpstart at the beginning of training compared to the other methods. We used RMSProp with momentum as the optimizer for image classification tasks.
>
> For the fifth review point, we are not sure what does the reviewer mean by “other relatedness method in reinforcement learning”. As mentioned in section 4, to the best of our knowledge, so far there has not been a working method that could quantify task relatedness in RL (Carroll and Seppi, 2005; Ammar et al., 2014). We would appreciate more suggestions from the reviewer on this. Regarding the reviewer’s question about noise in gradient, we have explained this in footnote 4 of our paper: “...we compute cosine similarity between a distillation gradient and a single sample of the policy gradient estimator, meaning that we are using a high variance estimate of the similarity...”. Specifically, in this RL experiment of distilling then transfer between the *same* task, the noise of the gradient in a single sample makes the tasks less similar---but they are actually the same (e.g., their cosine similarity should be very close to 1, but due to the noise, it could be much lower than 1 for a single sample).

---

### Official Review · AnonReviewer3 · 2018-11-02
**Interesting idea but weak experimental setup.**

**Rating:** 4
**Confidence:** 5

**Review:**

The paper is addressing the problem of a specific multi-task learning setup such that there are two tasks namely main task and auxiliary task. Auxiliary task is used for the sole purpose of helping the main one. In other words, auxiliary task performance is not of interest. The simple and sensible approach proposed in the paper is using cosine similarity between the gradients of two loss functions and incorporating the auxiliary one if it is positively aligned with the main gradient. Authors suggest to further scale loss functions using the cosine similarity but it only experiments with the simpler case of binary decision of using both gradients or only the main one. Authors provide a convergence guarantee (without any convergence rate) by simply extending the convergence of gradient method.

The paper is definitely addressing an important problem as the authors cite many previous work which uses the setup of set of auxiliary tasks helping a main one. The method is simple and easy to implement. Hence, it has a potential to be useful for the community.

One major issue for me is the experimental setup. The authors cite many interesting, realistic and practical setups (Zhang et al., 2016; Jaderberg et al., 2017; Mirowski et al., 2017; Papoudakis et al., 2018), but do not use any of these setups in their experiments. Instead, paper uses set of toy experiments. This is very puzzling to me as all these papers set existing baselines for interesting problems which authors can easily compare. I think the paper needs to be experimented and compared with these established methods.

Another major issue is the weak multi-task learning baseline used in the paper. There have been many interesting developments in adaptive scaling of multiple loss functions in the literature. However, paper does not compare with them. Example of these methods are: [GradNorm: Gradient Normalization for Adaptive Loss Balancing in Deep Multitask Networks, ICML 2018] and [Multi-Task Learning Using Uncertainty to Weigh Losses for Scene Geometry and Semantics, CVPR 2018]. Although these methods addresses the case of all tasks being important, it is a valid baseline and need to be compared. Similar to my first points, these papers also use very realistic and interesting experiments which would fit better than the toy experiments in the paper.

Final major issue is the fact that experimental results are suggesting the method is not effective. In ImageNet experiment, auxiliary tasks actually hurt the final performance as the single task is better than all methods including the proposed one. Proposed method does not guarantee that auxiliary tasks will have no harm. The GridWorld experiment is sort of a sanity check to me as it is very hand-crafted. For Breakout experiment, single task actually outperforms all baselines and this means the proposed method results in a harm similar to ImageNet case. For Breakout+MSPacMan experiment, multi task and the proposed method performs almost exactly same. I do not get why the performance on Breakout is relevant for this case since it is not a main task. The paper clearly states that only performance of an interest is the main one which is MSPacMan in this case. Also, in this experiment clearly all methods are still learning as the curve did not plateau yet. I am curious, why the learning is stop there. I do not think we need the method to be effective to be published; but, the negative result should be explained properly.

MINOR NITPICKS
- Algorithm 1&2 are crucial to understand the paper, they should be in main text
- ImageNet class IDs change between years. So, actual wordnet IDs or class names is a better thing to state
- What happens if there are multiple auxiliary tasks?
- Does the theory still hold for loss functions which are not Lipschitz as the Cauchy's gradient method requires that for convergence
In summary, the paper is proposing a sensible method for an important problem. However, it is only tested for toy problems although there are interesting existing setups which would be ideal for the method to be tested. Moreover, it is only compared with the most-naive multi task learning baselines. Even this limited experimental setup does not confirm what the paper is claiming (using auxiliary tasks only when they help). And the paper fails to explain this failure cases. The method needs to be experimented with a more realistic setup with more realistic baselines.

------
After rebuttal:

I gave detailed responses to each part of the rebuttal below. Here is the summary:

Although the response addresses some of my concerns. There are still major issues with the experimental study. 1) there are existing, relevant and well-studied multi-task setups with negative interference. Method should be experimented with some of those setups. 2) Multi-task baseline in the paper is naive and far from state-of-the-art. Paper need strong baselines as discussed. Hence, I am keeping my score. Paper needs to be improved with a stronger experimental study and need to be re-submitted.

---

> ### Author Response · Authors · 2018-11-27
> **Response to Reviewer 3: part 1**
>
> We thank Reviewer 3 for their detailed feedback!
>
> We have clarified our problem setup in Section 1.1: The goal is to devise an algorithm that can automatically (i) leverage the auxiliary task when it is helpful (e.g. learn faster) and (ii) block negative transfer when the auxiliary task is not helpful (i.e. recover the performance of training only on the main task). We have updated captions of figures to clarify our experimental setup.
>
> We address the major issues one by one.
>
> 1. “One major issue for me is the experimental setup.” We handpicked these tasks because they show rich behaviors in a way that *the auxiliary task is helpful to the main task initially, but hurt later on*---this is the setting we care about and our adaptive mechanism is reasonable under this scenario. In contrast, the established methods choose their auxiliary tasks to be the ones that always help. These tasks are usually simple, converge fast, and there is no significant negative interference that can happen during the training phase. For example, the auxiliary task in Mirowski et al. is a depth prediction task which predicts an extremely low dimensional version of the depth map, instead of trying to construct the full depth map---even though the latter might seems to be a more natural task. Our approach is meant to reduce the pressure on finding a suitable auxiliary task; if a task always helps, then simple multi-task could already work and there is no need for using our method to compute their gradient cosine similarities. We respectfully disagree that all of our experiments are “toy”. We experimented with ImageNet dataset, which even in the form of our modified binary task version is not an easy task. We also tested in the Atari domain, which is a difficult task for RL agents. Given the variety of our experimental designs (cross-domain and cross-task, supervised learning and reinforcement learning), we think it’s unfair to say that “paper uses set of toy experiments”.

---

> > ### Author Response · Authors · 2018-11-27
> > **Response to Reviewer 3: part 2**
> >
> > 2. “Another major issue is the weak multi-task learning baseline used in the paper.” We thank the reviewer for pointing out the recent developments in the related literature. However, in our opinion, our method is not directly comparable with the papers mentioned by the reviewer due to the following reasons (we have also cited these papers and added a discussion in the related work):
> >
> > (i) as the reviewer already noted, these papers aim to solve the multi-objective optimization problem (i.e., all tasks are of interest) which is not the same problem setup as our method. In particular for the GridWorld experiment, simply adding losses is actually the optimal thing to do in the multi-task setting  since there is no representation learning (policies are tabular). However, the simple adding method fails completely in our problem setup because of the fact that a multi-task solution is not what we are after---our goal is to extract useful information for a single important task. Similarly, for the example functions in Figure 1, any multi-task oriented adaptation scheme will fail because the minimum of the multi-task solution is nowhere near the solution of the main task. To summarize, multi-task techniques are not capable of making the right tradeoffs under the scenarios where one is only interested in the performance of the main task while all other tasks are just for the purpose of bootstrapping learning, and can potentially hurt performance on the main task.
> >
> > (ii) these papers proposed different ways of scaling the multi-loss function and they follow the intuition of finding a weight for each task individually (e.g., GradNorm scales the weight based on gradient magnitude of each task) without looking at their interaction. However, our method is not a “gradient scaling” method like in these papers; it is a vector field projection technique and this distinction is crucial to understand. Fundamentally, our proposed approach is a technique that allows adding additional gradient fields (or losses) in a way that guarantees converges to the local optima of the main task. The auxiliary vector fields, or gradients, can only affect the convergence speed (which we have empirically shown to be better), and we prove this convergence property in proposition 1 and 2 (see Appendix A.1 for details). The papers mentioned are trying to optimize a different goal. We will try to add experiments using the scaling methodologies as in these papers in the camera-ready version, but we emphasize that this would not be a fair comparison due to the distinct objectives between these methods and ours. Moreover, their mechanisms do not explicitly look for the relatedness of each task, while ours measures task relatedness using gradient cosine similarity.
> >
> > (iii) one should also note that the overall setting is different in these papers compared to ours. In the papers mentioned, they all have a multi-task setting where a neural network takes image inputs and produces multiple labels based on the *same input data*. In contrast, our setting considers *different input data* for each task. In ImageNet, we input images from two different classes; in RL GridWorld, the environments are of different distribution; in Atari games, Breakout and Ms. Pacman are two very distinct tasks. This difference is critical for the usefulness of our method versus their methods. For example, using gradient magnitude to scale loss function between Breakout and Ms. Pacman (as was done in GradNorm) would not work since their magnitude would be very different due to distinct input distributions and different environment dynamics in RL tasks. Furthermore, the tasks used in their papers (e.g. predicting segmentation map and depth from a single image) could inherently be helpful to each other (cosine similarity always greater than 0), so these tasks may not be the best to illustrate the problem our method is trying to solve.
> >
> > To summarize, these papers solve the problem of weighing individual losses for multi-task learning without looking at their interaction, whereas we solve the problem of deciding when to use auxiliary task when we ultimately only care about main task performance, by looking at gradient cosine similarity. Nonetheless, we agree that our approaches are complementary and could potentially be combined to further improve performance. For example, as the reviewer said, instead of using simply “binary decision of using both gradients or only the main one”, we could adapt ideas from existing literature to find a better weighing mechanism.

---

> > > ### Author Response · Authors · 2018-11-27
> > > **Response to Reviewer 3: part 3**
> > >
> > > 3. final major issue “experimental results are suggesting the method is not effective.”: Our goal was to show that our method can successfully detect and block negative transfer, while recovering the performance of multi-task learning on helpful auxiliary tasks; our experiments on supervised learning and reinforcement learning supports our claims that our method is effective at this.
> > >
> > > In ImageNet experiment of far class pair [48, 920], it shows exactly this property---when class 48 detects that the loss from class 920 is negatively affecting its learning, it ignores class 920 and continue its learning, thus in the end matching the single task performance (the performance boost after around 15k steps). While in the naive multi-task setting, class 48 is not able to gain this boost because of the consistent negative effect from class 920. Similarly for near class pair [871, 484] that our method is comparable to multi-task learning because both losses share similar gradient direction and there is no dropping occurs. We have added a description in the caption of Figure 2 to clarify this in our paper.
> > >
> > > In Atari experiment of Breakout and Ms. Pacman, we’d like to clarify what the reviewer said about “...I do not get why the performance of breakout is relevant...”. The performance of Breakout is relevant because in this setting, our main task is to learn Breakout and Ms. Pacman simultaneously (i.e. the main task is multi-task). We chose the main task itself to be multi-task to illustrate a complex scenario where auxiliary task helps with only part of the main task, and that too only initially.  Therefore, we are not comparing “single task vs. multi-task”, but to use an auxiliary task, a teacher agent with distilled policy from Breakout only,  to bootstrap the learning of both Breakout and Ms. Pacman. In Figure 5 of our paper we show that, when always distilling from the teacher (Multitask RL + distillation)  the agent learns only Breakout but not Ms. Pacman, because the teacher only knows about Breakout; when never distilling (Multitask), the reward from one game saturate the learning of another. As shown in the “Normalized Average Score” plot of Figure 5, our method performs better than all these methods because the agent is able to learn both games well.
> > >
> > > Regarding the reviewer’s concern that the Ms. Pacman’s learning curve was not plateaud yet, we’d like to clarify that we followed a typical setting on Atari where one usually set a fixed “budget” of steps and compare performance at that point in time. This is a reasonable setting since 1) our method cares about bootstrap learning and 2) early stopping is not well defined in RL. We could increase the budget and train longer, but that will be more resource-consuming and will not change the current presented results in any meaningful way.
> > >
> > > ---
> > >
> > > To summarize, our goal was to propose a method that automatically (i) leverages the auxiliary task when it is helpful (e.g. learn faster) and (ii) block negative transfer when the auxiliary task is not helpful. Our experiments support our main achieves this and shows that our method can recover multi-task performance when auxiliary task helps and gracefully recover performance of single task when auxiliary task hurts. We have demonstrated that our method works well on a variety of supervised learning as well as reinforcement learning experiments (which are more challenging). In addition, we have noted and shown in Appendix D an example of where our method could slow down convergence and discussed a few drawbacks.
> > >
> > > We hope we have addressed your major concerns.
> > >
> > > “MINOR NITPICKS”
> > > - We have revised our paper to include Algorithm 1 in the main text.
> > > - We have added a description for each class ID in the main text
> > > - Scaling our method to many tasks should be relatively straightforward since the treatment of each auxiliary loss is done independently (as the gradient of each auxiliary loss is projected on the gradient of the task of interest). We have added this as a future work in discussion.
> > > - Regarding “Does the theory still hold for loss functions which are not Lipschitz as the Cauchy's gradient method requires that for convergence”, this is a good point. We will look into this in future work.

---

> > > > ### Comment · AnonReviewer3 · 2018-11-27
> > > > **Partially agree with the response**
> > > >
> > > > If the main purpose of the experiments is showing that no negative interference happens during training, I partially agree with the authors since it is showing better performance than naive multi-task learning with equal weighting. On the other hand, in ImageNet the single task outperforms both methods; hence, some negative interference happens. Claiming it as a success is little bit far fetched. Only conclusion I see is that the method has less negative interference than most naive multi-task baseline.
> > > >
> > > > For learning Breakout and Ms. Pacman simultaneously comment, I misunderstood that part of the paper. Thanks for the clarification.

---

> > > ### Comment · AnonReviewer3 · 2018-11-27
> > > **Disagree with the response**
> > >
> > > (i) I strongly disagree with the authors that they are not fair baselines. Authors already compare with multi-task learning. However, they choose the most naive multi-task learning algorithm in the literature (equal weighting). A fair comparison to multi-task learning requires comparison to strong multi-task learning baselines. All the cited papers are multi-task learning papers and authors need to choose a strong multi-task learning baseline. Authors use sentences like "...we compare single-task training, multi-task training...". These claims require strong multi-task baselines not naive baselines.
> > >
> > > (ii)/(iii) Yes, they are doing something completely different since they are doing multi-task learning. But, paper is claiming the proposed method is better than multi-task learning. Hence, paper need a strong multi-task learning baselines.

---

> > > > ### Author Response · Authors · 2018-11-28
> > > > **Clarifying what we mean by "multi-task"**
> > > >
> > > > We suspect that this disagreement is primarily caused by what we mean by “multi-task learning” versus what the reviewer means by “multi-task learning”.
> > > >
> > > > To be more specific, we further clarify the differences between [Chen et al,. ] & [Kendall et al.,] and our paper, assume we have two tasks, L_{main} and L_{aux}:
> > > >
> > > > [Chen et al,. ] & [Kendall et al.,]:
> > > > - Problem setup: achieve good performance on both L_{main} and L_{aux}
> > > > - They optimize [ w_{main} * L_{main} + w_{aux} * L_{aux} ], where w_{main} is tuned based on L_{main} only; w_{aux} is tuned based on L_{aux} only.
> > > >
> > > > Our paper:
> > > > - Problem setup: achieve good performance on L_{main} only
> > > > - We optimize [ L_{main} + \lambda * (L_{aux} ], where \lambda = 2*sign(cosine( \nabla_L_{main}, \nabla_L_{aux} ))-1, we refer to this as "our method"
> > > > - NOTE: *\lambda depends on both L_{main} and L_{aux}. This is the crucial difference*
> > > >
> > > > Since \lambda is a binary weight in our “unweighted” version, we compare to:
> > > > (i) L_{main}, which is equivalent to always set \lambda=0, we refer to this as “single task”
> > > > (ii) L_{main} + L_{aux}, which is equivalent to always set \lambda=1, we refer to this as “multi-task”
> > > > For this reason (that the weight \lambda is binary), we believe this is a fair comparison to test the effectiveness of our method without additional confounding factors. We did not intend to claim our method is better than all variants of multi-task learning papers such as those in [Chen et al,. ] & [Kendall et al.,] (they are solving a different problem, see also the earlier comment). Apologies if this was unclear, we will be happy to rephrase the claims/text in our paper.
> > > >
> > > > As we mentioned earlier, the work of [Chen et al,. ] & [Kendall et al.,] is complementary as they address the problem of scaling losses individually, while our method focuses on capturing the alignment between losses, therefore, deciding if an auxiliary task will be helpful for the main task and for how long. One way to combine the strengths of our method and their work would be to optimize the following:
> > > > w_{main} * L_{main} + \lambda * w_{aux} * L_{aux}
> > > > We leave this for future work.
> > > >
> > > > Directly comparing our method to L_{main} + w_{aux} * L_{aux} would not be appropriate as
> > > > (i) it is a bit unfair to those methods as w_{aux} only depends on L_{aux}
> > > > (ii) due to the confounding factors (aux loss is “weighted” in one vs “unweighted” in the other), it is hard to draw any meaningful conclusions about whether the performance difference is coming due to individually scaling the loss or scaling the loss by the alignment.

---

> > ### Comment · AnonReviewer3 · 2018-11-27
> > **Misunderstandings/Errors**
> >
> > The statement that "In contrast, the established methods choose their auxiliary tasks to be the ones that always help" is simply wrong. All the loss scaling methods actually show that naive implementation of the multi-task learning actually hurts some times. For example, [Chen et al., Table 2]: naive multi-task learning hurts normal estimation error. [Kendall et al., Table 1]: naive multi-task learning hurts segmentation performance. In summary, these experiments are still very interesting for the practitioners and proposed method is applicable. And, the statement that "there is no significant negative interference" is NOT correct.
> >
> > I think authors misunderstood my definition of toy problem. I did not mean an easy problem. The paper uses set of problems which are not realistic and practical. They are designed to specifically satisfy the assumptions of the setup in the paper. Although they are scientifically valuable, they are not relevant to the practitioners. As I said, the existing problems studied in the literature already shows significant negative interference and relevant to the practitioners. So, I see no reason for them to be included in the paper.

---

> > > ### Author Response · Authors · 2018-11-28
> > > **Clarification**
> > >
> > > Apologies for not being clearer: we meant the statement “established methods … ” in the context of usual auxiliary tasks in RL, i.e., [Jaderberg et al.,] and [Mirowski et al.,]. As we have pointed out in the previous rebuttal, “the auxiliary task in Mirowski et al. is a depth prediction task which predicts an extremely low dimensional version of the depth map, instead of trying to construct the full depth map”. Similarly in Jaderberg et al., 2017, an example is the use of an “immediate reward prediction” as the auxiliary task, which explicitly helps the agent in shaping its internal feature learning thus more efficient in terms of the amount of experience needed to learn.
> > >
> > > This is very different from the setup in [Chen et al], [Kendall et al.], so we agree that our statement wouldn’t be applicable in those contexts. While it is true that in their papers equal weight hurts the performance in one task or another, it is not clear that the results in [Chen et al., Table 2] and [Kendall et al., Table 1] are caused necessarily by negative interference. A confounding factor in their paper is that the scales/units of their losses are very different. The fact that their proposed solutions, which take into account only the scale of the losses without looking at their interaction, can still achieve better performance, shows that part of the issue is ensuring that the individual losses are on the same scale (through the course of training) and not that one necessarily hurts the performance of the other. Quoting [Kendall et al.], their method “allows us to simultaneously learn various quantities with different units or scales in both classification and regression settings”, quoting [Chen et al] their goal is to “place gradient norms for different tasks on a common scale through which we can reason about their relative magnitudes”.  These methods are primarily addressing the scaling problem and not directly addressing negative interference as we are (see also below where we further clarify the differences).
> > >
> > > We disagree with the reviewer that the "paper uses set of problems which are not realistic and practical". For supervised learning, we use (subsets of) ImageNet dataset which is more challenging than say MNIST/SVHN used in publications. For RL, Atari games have been extensively used as the benchmark in many publications. From both perspectives, we believe our experiment design is realistic and valid, and comparable with other benchmarks used in the literature.
> > >
> > > Regarding the comment that our experiments were “designed to satisfy the assumptions of the setup in the paper”. We picked these experiments as we know the ground truth of whether an auxiliary task helps or not, and we can scientifically validate whether our experiments agree with our hypothesis. Note that our methods are not given any kind of privileged information (whether the auxiliary loss helps or hurts) during training, the ground truth is just for us to verify if our method is working as intended. We believe it is completely reasonable to define particular tasks and in fact preferable to make such assumptions explicit, as it allows us to show that the method indeed works due to the reasons cited in the paper.
> > >
> > > As mentioned earlier, we’re happy to additionally include experiments using the multi-task setup of [Kendall et al.] or [Chen et al] in the final version. To the best of our knowledge, the ground truth task similarity is not available and the code for neither of those papers is publicly available (we’d appreciate pointers if the reviewer is aware of an implementation!), so we need to reproduce their experimental setup from scratch, which was not possible in the short rebuttal time. We’ll include this experiment in the final version.
> > >
> > > "Although they are scientifically valuable, they are not relevant to the practitioners.": We believe our work is relevant to practitioners and valuable to the community. See also Reviewer 2‘s comments which said “the proposed method is a welcome addition to the tool box of practitioners.”

---

### Author Response · Authors · 2018-11-27
**General Response**

We thank all reviewers for their detailed feedback. We have uploaded a revised version addressing all of the concerns raised by the reviewers. Briefly, highlights of the major changes we made include:
- Moved Algorithm 1 from Appendix C to Section 3;
- Added ImageNet class names in Section 3.1;
- Cited and discussed the papers mentioned by Reviewer 3 in Section 4;
- Text revisions to improve readability.
We also address individual reviewer comments below.

---

### Meta-Review · Area_Chair1 · 2018-12-14
**Further validation of algorithm needed**

**Confidence:** 5
**Recommendation:** Reject

**Metareview:**

This paper tackles the problem of using auxiliary losses to help regularize and aid the learning of a "goal" task. The approach proposes avoiding the learning of irrelevant or contradictory details from the auxiliary task at the expense of the "goal" tasks by observing cosine similarity between the auxiliary and main tasks and ignore those gradients which are too dissimilar.

To justify such a setup one must first show that such negative interference occurs in practice, warranting explicit attention. Then one must show that their algorithm effectively mitigates this interference and at the same time provides some useful signal in combination with the main learning objective.

During the review process there was a significant discussion as to whether the proposed approach sufficiently justified its need and usefulness as defined above. One major point of contention is whether to compare against the multi-task literature. The authors claim that prior multi-task learning literature is out of scope of this work since their goal is not to measure performance on all tasks used during learning. However, this claim does not invalidate the reviewer's request for comparison against multi-task learning work. In fact, the authors *should* verify that their method outperforms state-of-the-art multi-task learning methods. Not because they too are studying performance across all tasks, but because their method which knows to prioritize one task during training should certainly outperform the learning paradigms which have no special preference to one of the tasks.

A main issue with the current draft centers around the usefulness of the proposed algorithm. First, whether the gradient co-sine similarity is a necessary condition to avoid negative interference and 2) to show at least empirically that auxiliary losses do offer improved performance over optimizing the goal task alone. Based on the experiments now available the answers to these questions remains unclear and thus the paper is not yet recommended for publication.